# High PD-L1 Expression Correlates with an Immunosuppressive Tumour Immune Microenvironment and Worse Prognosis in *ALK*-Rearranged Non-Small Cell Lung Cancer

**DOI:** 10.3390/biom13060991

**Published:** 2023-06-15

**Authors:** Xia Tian, Yalun Li, Qin Huang, Hao Zeng, Qi Wei, Panwen Tian

**Affiliations:** Department of Pulmonary and Critical Care Medicine, Lung Cancer Center, West China Hospital, Sichuan University, Precision Medicine Key Laboratory of Sichuan Province, Chengdu 610041, China; 2020224025256@stu.scu.edu.cn (X.T.); lunlunlee@163.com (Y.L.); aqin2019up@163.com (Q.H.); zenghao5577aak@163.com (H.Z.); weiqi8602@163.com (Q.W.)

**Keywords:** *ALK*, PD-L1, tumour-immune microenvironment, multiplex immunofluorescence, non-small cell lung cancer

## Abstract

High tumour programmed cell death-ligand 1 (PD-L1) expression is associated with poor progression-free survival (PFS) after tyrosine kinase inhibitor (TKI) therapy in *ALK*-rearranged non-small cell lung cancer (NSCLC). However, the characteristics of the tumour microenvironment (TME) and their prognostic values in *ALK*-rearranged NSCLC are unknown. Here, we collected tumour tissues from pretreated *ALK*-rearranged NSCLC patients, immunohistochemical staining was used to assess PD-L1 expression, and tumour-infiltrating immune cells were determined via multiplex immunofluorescence staining (mIF). Our data showed that the median values of PFS for the high PD-L1 group and low PD-L1 group who received ALK-TKI treatment were 4.4 and 16.4 months, respectively (*p* = 0.008). The median overall survival (OS) of the two groups was 24.0 months and not reached, respectively (*p* = 0.021). Via univariate and multivariate analyses, a high PD-L1 expression and a worse ECOG PS were determined to be independent prognostic factors of OS (HR = 3.35, 95% CI: 1.23–9.11, *p* = 0.018; HR = 6.42, 95% CI: 1.45–28.44, *p* = 0.014, respectively). In addition, the high PD-L1 group had increased Tregs and exhausted CD8+ T cells in both the tumour and stroma (all *p* < 0.05). High PD-L1 expression was an adverse predictive and prognostic biomarker for *ALK*-rearranged NSCLC. The characteristics of the TME in patients with high PD-L1 expression were shown to have an immunosuppressive status.

## 1. Introduction

Lung cancer is currently the main leading cause of cancer-related deaths worldwide [1]. Non-small cell lung cancer (NSCLC) comprises approximately 85% of lung cancers overall [2]. Approximately 3–7% of patients with NSCLC harbour rearrangements in the anaplastic lymphoma kinase (*ALK*) gene, a potent oncogenic driver [3], and ALK tyrosine kinase inhibitors (TKIs) have markedly improved the survival of these patients. However, not all patients with *ALK*-rearranged NSCLC can benefit from ALK-TKIs, and it is important to identify these patients to improve their survival outcomes [4].

Programmed cell death-ligand 1 (PD-L1) in tumours was identified as a positive predictive biomarker for advanced NSCLC patients treated with immune checkpoint inhibitors (ICIs) [5]. PD-L1 expressed on tumour cells interacts with programmed cell death protein 1 (PD-1), which is expressed on cytotoxic T cells to suppress T cell activation and growth, and negatively regulates the antitumour immune response. Several studies have shown that high tumour PD-L1 expression is linked to worse progression-free survival (PFS) in response to ALK-TKIs [6,7,8,9,10]. However, the association between PD-L1 expression and prognosis in *ALK*-rearranged NSCLC has not been well studied. A prior study showed that the status of PD-L1 expression alone did not influence the OS of *ALK*-positive NSCLC patients [11]. However, another recent study found that high baseline PD-L1 expression was associated with shorter overall survival (OS) in *ALK*-rearranged lung adenocarcinoma [10]. Hence, the relationship between PD-L1 expression and prognosis in *ALK*-rearranged NSCLC remains unclear.

Previous have studies stratified the tumour microenvironment (TME) based on tumour-infiltrating lymphocytes (TILs) and PD-L1 expression, in which the PD-L1+TILs+ TME is most likely to benefit from ICI therapy [12,13]. Mazzaschi et al. observed that resected NSCLC patients with high levels of CD8+ lymphocytes lacking PD-1 had a longer OS [14]. Yang et al. reported that tumour PD-L1 expression and increased CD4+ T cell infiltration were independent predictors of better OS in SCLC [15]. Based on this evidence, it appears that the TME is associated with prognosis. However, the association between the TME and prognosis in *ALK*-rearranged NSCLC remains poorly understood. Yang et al. performed multiplex immunofluorescence staining (mIF) for three cases, and found that cases with high PD-L1 expression (PD-L1 TPS 80%) showed less infiltration of CD20+ B cells, CD8+ T cells, and CD4+ T cells [8]. Most *ALK*-rearranged NSCLC with high tumour PD-L1 expression (TPS ≥ 50%) seems to lack high levels of TILs [12,16]. However, the difference in the TME between patients with high and low PD-L1 expression is unclear. Hence, there is a lack of studies evaluating the TME of tumours with high PD-L1 expression in *ALK*-rearranged NSCLC and the impact of the TME on the prognosis of these patients.

Therefore, we evaluated the association between baseline PD-L1 expression and prognosis in patients with *ALK*-rearranged advanced NSCLC, performed a comprehensive analysis of the TME of patients with a high PD-L1 expression, and analysed the impact of the TME on prognosis.

## 2. Materials and Methods

### 2.1. Patients

We reviewed the medical records of 4261 consecutive patients diagnosed with stage IIIB-IVB NSCLC who were tested for *ALK* rearrangement and PD-L1 expression using immunohistochemical staining at the West China Hospital between December 2014 and August 2020.

The inclusion criteria were as follows: (1) histologically confirmed unresectable stage IIIB-IVB NSCLC (eighth TNM classification) [17]; (2) *ALK* rearrangement; (3) available data on clinical response to ALK-TKI therapy; and (4) adequate pretreatment tissue specimens for multiplex immunofluorescence staining (mIF). The exclusion criteria were as follows: (1) lack of PD-L1 expression data for tumour specimens and (2) undetailed treatment information or lost for follow-up. Finally, 52 *ALK*-rearranged advanced NSCLC patients were included, 18 of whom had sufficient specimens available for mIF (Figure 1). 

### 2.2. Immunohistochemical Staining of ALK

Tumour *ALK* status was determined via immunohistochemistry (IHC) staining using the ALK (D5F3) assay with a Ventana BenchMark XT automated staining instrument (Ventana, AZ, USA), according to the manufacturer’s instructions.

### 2.3. mIF

mIF was performed on sequential 4 μm thick sections obtained from FFPE lung cancer specimens by using a PANO 7-plex IHC kit (cat 0004100100) (Panovue, Beijing, China). We grouped the immunofluorescence markers used into two six-antibody panels as follows: Panel 1 comprised PANCK, CD4, CD8, Foxp3, PD-1 and DAPI; and Panel 2 comprised PANCK, CD3, CD20, CD56, CD68 and DAPI. Based on the marker expression and density features of the cells, eight distinct major populations were identified, namely, CD4 T cells (CD4+), CD8 T cells (CD8+), Tregs (CD4+FOXP3+), exhausted CD8 T cells (CD8+PD-1+), CD3 T cells (CD3+), B cells (CD20+), NK cells (CD56+) and macrophages (CD68+) [18]. Briefly, the slides were deparaffinised, rehydrated, and subjected to antigen retrieval by placing them in a plastic jar filled with antigen retrieval solution (citric acid solution, pH 6.0/pH 9.0) and heating them in a microwave for 45 s at 100% power followed by 15 min at 20% power. Tissues were incubated with blocking solution for 10 min. Only one antigen was detected in each round, which included primary antibody incubation, horseradish peroxidase-conjugated secondary antibody incubation, and fluorescent labelling using tyramide signal amplification, followed by labelling with the next antibody after epitope retrieval and protein blocking, as performed previously. After the last round of antibody staining, all slides were counterstained with 4′-6′-diamidino-2-phenylindole (DAPI; D9542, Sigma–Aldrich, St. Louis, MO, USA).

### 2.4. Tissue Imaging and Analysis

The stained slides were scanned using the Mantra System (PerkinElmer, Waltham, MA, USA). Multispectral images were separated using a spectral library established from a single stained tissue image for each reagent using inForm software (PerkinElmer, Waltham, MA, USA). For batch analysis, inForm software can automatically detect and segment specific tissue compartments into the tumour parenchyma and stroma based on the epithelial cell marker pancytokeratin and improve phenotyping using DAPI-based cell segmentation. The intratumour region was defined as malignant cell nests, and the stroma was characterised by the fibrous tissue present between malignant cells. For each staining marker, a positive threshold within the nucleus or cytoplasm was set, and the entire image set was analysed. The generated data included the count, density and percentage of positive cells for each fluorescent marker. The single-cell-level data exported from inForm were further processed using the R statistical programming language (Version 3.1). Using Olympus, 2–3 regions of interest (ROIs) per slide were selected when each tumour tissue was scanned using low magnification (4×). The invasive margin (IM) was defined as the area within 250 μm on each side of the border separating the host tissue from malignant tissue, and then two distinct regions were obtained: IM tumour (IM-T) and IM stroma (IM-S). 

### 2.5. Evaluation of Immune Cell Spatial Distribution

To evaluate the spatial relationships between immune cells and tumour cells, we conducted spatial analysis, including calculating the average number of immune cells distributed within a 30 μm radius from the nuclear centre of any given tumour cell and using “nearest neighbour distance” (NND) analysis; this was performed by calculating the distance (μm) from each tumour cell to its nearest neighbour immune cell. We selected a radius of 30 μm for study, as it has been previously suggested that this is a reasonable physiologic distance to identify immune cells that may be able to engage in direct and cell-to-cell interactions with tumour cells [19].

### 2.6. Treatment Evaluation

Clinical response to ALK-TKIs was evaluated based on a computed tomography (CT) scan follow-up every 3 months during treatment according to the Response Evaluation Criteria in Solid tumours, version 1.1 (RECIST 1.1) [20]. PFS was defined as the period from the start of ALK-TKI treatment to disease progression or death. OS was defined as the interval between diagnosis and death or the last follow-up. The objective response rate (ORR) was defined as the percentage of patients achieving complete response (CR) or partial response (PR). The disease control rate (DCR) was defined as the percentage of patients achieving CR, PR or stable disease (SD).

### 2.7. Statistical Analysis

The chi-squared test was used to analyse ORR and DCR according to PD-L1 expression. Kaplan–Meier curves and the log-rank test were used to analyse PFS and OS according to PD-L1 expression and to provide median survival times based on immune cell exposures. The prognostic factors of PFS and OS were analysed via univariate and multivariate Cox regression analyses. Variables with *p* < 0.05 in the univariate analysis were included in the multivariate Cox regression model. The hazard ratio (HR) and 95% confidence interval (CI) were calculated for all variables in the regression model. The Mann–Whitney U test was used to examine the composition, density, location and spatial distribution of immune cells according to PD-L1 expression. The Spearman correlation test was used to assess the correlations between immune cells. The value of the coefficient ranged from −1 to 1, and an absolute value of the correlation coefficient of “>0.60” was defined as a strong correlation in our study. The optimal cut-off values of the immune cell percentage were obtained using the surv_cutpoint function. All tests were two sided, and statistical significance was set at *p* < 0.05. All statistical analyses were performed using SPSS Statistics (version 25.0), GraphPad Prism (version 9.0) and R version 4.1.2.

## 3. Results

### 3.1. Patient Characteristics 

A total of 52 patients with *ALK*-rearranged lung adenocarcinoma were enrolled in this study. The baseline characteristics of the patients are shown in Table 1. The median age was 51 (range, 41 to 61) years, 60% (31/52) were females, 73% (38/52) were non-smokers, and 94% (49/52) were diagnosed with stage IV disease. Thirty-five percent (18/52) had an ECOG PS of 0, 52% (27/52) had an ECOG PS of 1, and 13% (7/52) had an ECOG PS of 2. Forty-four percent (23/52) had a PD-L1 TPS of 0%, 33% (17/52) had a PD-L1 TPS of 1–49%, and 23% (12/52) had a PD-L1 TPS of ≥50%. Forty-two patients received crizotinib, and 10 patients received alectinib. All patients received ALK-TKIs as first-line therapy. 

Within the whole cohort, 40 patients were assigned to the PD-L1 low group (PD-L1lo group), and 12 patients were assigned to the PD-L1 high group (PD-L1hi group). There was no significant difference between the clinical characteristics of the two groups (Table 2).

### 3.2. Clinical Outcomes in Patients with Different PD-L1 Expression Levels

The median follow-up time was 37.3 months (20.9–50.7 months). The ORR was 70% for the PD-L1lo group and 50% for the PD-L1hi group (*p* = 0.202), while the DCR was 97.5% for the PD-L1lo group and 83.3% for the PD-L1hi group (*p* = 0.065) (Figure 2A,B). The median PFS was significantly shorter in the PD-L1hi group than in the PD-L1lo group (4.4 vs. 16.4 months, *p* = 0.008) (Figure 2C). The median OS for patients with high PD-L1 expression was 24.0 months (95% CI: 19.3 months-NR), whereas the median OS was not reached (95% CI: 52 months-NR) for patients with low PD-L1 expression (*p* = 0.021) (Figure 2D). The univariate and multivariate analyses showed that high PD-L1 expression and bone metastasis were independent prognostic factors of PFS (HR = 2.79, 95% CI: 1.35–5.79, *p* = 0.006; HR = 2.12, 95% CI: 1.12–4.03, *p* = 0.021, respectively) (Table 3), and high PD-L1 expression and a worse ECOG PS were independent prognostic factors of OS (HR = 3.35, 95% CI: 1.23–9.11, *p* = 0.018; HR = 6.42, 95% CI: 1.45–28.44, *p* = 0.014, respectively) (Table 4).

### 3.3. Features of the Tumour Immune Microenvironment in Patients with High PD-L1 Expression

Of the 52 patients, 18 patients (10 PD-L1hi and 8 PD-L1lo) had sufficient tissue available for mIF analysis (Figure 3A,B). The percentages of Tregs and exhausted CD8+ T cells in the PD-L1hi group were higher than those in the PD-L1lo group (2.55 × 10^−3^ vs. 6.60 × 10^−5^, *p* = 0.005; 1.31 × 10^−3^ vs. 1.65 × 10^−4^, *p* = 0.005, respectively), while the percentages of CD3+ T cells, B cells, NK cells and macrophages were not significantly different between the two groups (all *p* > 0.05) (Figure 3C–J). The segmentation of tumours from the tumour nest and stroma showed that the percentages of Treg and exhausted CD8+ T cells were higher in the PD-L1hi group than in the PD-L1lo group in both the tumour and stroma (all *p* < 0.05) (Figure 3K–L). The stromal Treg/CD8 ratio was higher in patients with high PD-L1 expression than in patients with low PD-L1 expression (12.0% vs. 0.01%, *p* = 0.015) (Figure 3M).

No significant difference was observed in the percentages of Treg or exhausted CD8+ T cells between the PD-L1hi group and PD-L1lo group in the invasive margin (all *p* > 0.05, Figure 4A,B). We performed an NND analysis measuring the distance from each tumour cell to the closest Treg and exhausted CD8+ T cells (Figure 4C) and calculated the numbers of Treg and exhausted CD8+ T cells within a 30 μm radius of tumour cells (Figure 4D). Fourteen patients (8 PD-L1hi and 6 PD-L1lo) had computable Treg and exhausted CD8+ T cells in NND analysis, and Tregs and exhausted CD8+ T cells in the PD-L1hi group were located closer to tumour cells than those in the PD-L1lo group, but the difference was not statistically significant (*p* = 0.181 and 0.081, respectively) (Figure 4E,F). However, the numbers of Tregs and exhausted CD8+ T cells around tumour cells within a 30 μm range were significantly higher in the PD-L1hi group than in the PD-L1lo group (4.55 × 10^−2^ vs. 2.33 × 10^−4^, *p* = 0.003; 1.04 × 10^−2^ vs. 0, *p* = 0.001, respectively) (Figure 4G,H). In the PD-L1hi group, we found a strong correlation between intratumoural exhausted CD8+ T cells and stromal B cells (r = −0.762, *p* = 0.037) (Figure 4I). In addition, a strong correlation between intratumoural exhausted CD8+ T cells and intratumoural B cells was also observed (r = −0.833, *p* = 0.015) (Figure 4J).

### 3.4. PFS and OS of ALK-Rearranged NSCLC Based on TME

Of the 18 patients, 16 patients received crizotinib, and 2 patients received alectinib. To evaluate the clinical response of the patients to ALK-TKIs based on TME status, we performed survival analysis in patients treated with crizotinib. Kaplan–Meier curves revealed that patients with a high Treg level had a shorter PFS and OS than those with a low Treg level (9.8 vs. 35.5 months, *p* = 0.008; 24.0 months vs. NR, *p* = 0.007, respectively) (Figure 5A,C). In addition, patients with higher exhausted CD8+ T cell levels had a worse PFS than those with low exhausted CD8+ T cell levels (10.2 vs. 51.1 months, *p* = 0.005) (Figure 5B,D).

High Treg levels were present in 71.4% (5/7) of patients within the PD-L1hi group and in 22.2% (2/9) of patients within the PD-L1lo group. In addition, 42.9% (3/7) of cases belonging to the PD-L1hi group showed a high level of exhausted CD8+ T cells compared with 33.3% (3/9) of the PD-L1lo group. Our data revealed that patients with low exhausted CD8+ T cell levels showed trends towards an association with a longer PFS than patients with high exhausted CD8+ T cell levels in the PD-L1lo group (26.4 vs. 51.1 months, *p* = 0.077) (Figure 6C). Although not reaching a statistically significant impact on OS (Appendix A), we found that patients with a high PD-L1 expression, low Treg levels and high exhausted CD8+ T cell levels exhibited the longest OS, at 51.4 months (Appendix A).

## 4. Discussion

Our study demonstrated that high PD-L1 expression was independently associated with poor prognosis in *ALK*-rearranged patients receiving ALK-TKI treatment. Patients with high PD-L1 expression had a large number of Tregs and exhausted CD8+ T cells in the TME, which was associated with prognosis in *ALK*-rearranged patients.

We found that high PD-L1 expression was associated with a worse OS in *ALK*-rearranged NSCLC patients. Recently, accumulating evidence has suggested that PD-L1 expression could impact the efficacy of ALK-TKIs in *ALK*-rearranged NSCLC [8,9]. However, the impact of PD-L1 expression on the prognosis of ALK-TKIs remains controversial. A prior study showed that the status of PD-L1 expression alone did not influence the OS of *ALK*-positive NSCLC patients [11]. Zhou et al. observed that patients with a high baseline PD-L1 expression status (TPS ≥ 50%) had a significantly shorter median OS after crizotinib treatment [10]. This difference could be due to the number of cases analysed in the studies. In our study, we found that high PD-L1 expression was identified as an independent prognostic biomarker, wherein patients with a high PD-L1 expression had a worse OS. We speculate that an immunosuppressed TME may contribute to the poorer prognosis of patients with high PD-L1 expression. The abnormal activation of the PD-1/PD-Ll signalling pathway can inhibit the proliferation and differentiation of T cells and induce their apoptosis in a variety of different ways, thereby mediating the immune escape of tumours and disease progression. A previous study showed that PD-L1 upregulation is correlated with the dysfunction of tumour-infiltrating CD8+ T lymphocytes in NSCLC patients [21]. Such evidence suggests that immunological factors could play an important role in patients’ response to ALK-TKIs. It is essential to explore the differences in the TME of *ALK*-rearranged NSCLC with high and low PD-L1 expression.

We found that the number of Tregs in patients with high PD-L1 expression was higher than that in patients with low PD-L1 expression. The association between PD-L1 expression and Treg infiltration has been shown in several cancers [22,23,24]. However, it has not been reported in lung cancer. Similar to other studies, we demonstrated that tumour-associated PD-L1 expression correlated with increased Treg infiltration in *ALK*-rearranged NSCLC. These results were supported by the findings that PD-L1 could increase Foxp3 expression and that PD-L1 could convert naïve T cells to the iTreg phenotype [25,26]. Previous studies reported that the TME of *ALK*-positive NSCLC was immunosuppressive with a high population of immunosuppressive cells, such as Tregs, and a high expression of immunosuppressive markers [27,28]. Budczies et al. compared the immune gene expression profile and specific immune cell population levels of *ALK*-positive and *ALK*-negative lung adenocarcinoma, and found that the proportion of Tregs increased significantly in *ALK*-positive tumours [28]. Consistent with prior studies, our data indicated that there is a large number of Tregs in the TME of *ALK*-rearranged NSCLC patients with high PD-L1 expression. A previous study showed that Tregs are immunosuppressive cells that play an important role in shaping the antitumour immune response [29]. We found that the number of Tregs around tumour cells within a 30 μm range was higher in the PD-L1hi group than in the PD-L1lo group. These results indicated that the immunosuppressive function of Tregs in the TME was probably based not only on their number, but also on their spatial distribution.

In our study, *ALK*-rearranged NSCLC patients with high PD-L1 expression had increased exhausted CD8+ T cell levels. The percentage of exhausted CD8+ T cells around tumour cells within the 30 μm range in PD-L1hi patients was also higher. Our findings were in line with a previous study showing that the higher tumour cell expression of PD-L1 was correlated with a higher density of PD-1+ tumour-associated T cells [30]. These findings were supported by the work of Pardoll et al., who showed that activated PD-L1 expression in tumour cells could promote CD8+ T cell exhaustion in the TME [31]. A previous study reported that a significant number of PD-1-positive CD8+ T cells infiltrated the *ALK*-positive tumour bed [32]. The combined analysis of PD-L1 and CD8+ TILs showed a significantly higher proportion of PD-L1-/TIL- tumours and a lower proportion of PD-L1+/TIL+ tumours in *ALK*-positive patients than in *ALK*-negative patients [33]. The upregulation of PD-1 expression was associated with tumour antigen-specific CD8+ T cell dysfunction in NSCLC, and PD-1+ CD8+ T cells could result in immunosuppression in cancer [18,21]. These results suggested that the TME in *ALK*-rearranged NSCLC patients with high PD-L1 expression may be inhibited.

In our study, compared with the PD-L1lo group, the PD-L1hi group showed that Tregs and exhausted CD8+ T cells were closer to tumour cells, but this difference was not statistically significant. A multitude of studies have indicated that the proximity measurement of specific immune cells may be a potential indicator for survival [34,35]. Our results suggested that the spatial relationships between Tregs or exhausted CD8+ T cells and tumour cells may not contribute to survival, which is inconsistent with the outcomes reported in previous studies [36,37]. A possible explanation for this might be that our study was limited by a relatively small sample size. The precise relationships between Tregs and tumour cells and between exhausted CD8+ T cells and tumour cells in the microenvironment of *ALK*-rearranged NSCLC require further study.

We found that high Treg levels and high exhausted CD8+ T cell levels were associated with a worse PFS and OS. A recent study analysed advanced NSCLC biopsies obtained at different TKI treatment time points and found that regulatory T cells and CD8+ T cells expressing PD-1 were enriched at the PD time point [38]. Studies in NSCLC showed that patients with CD8 lymphocytes lacking PD-1 were associated with a more favourable clinical outcome than those harbouring CD8 lymphocytes expressing PD-1 [14,39]. Tregs can secrete TGF-β, IL-10, and IL-35, which downregulate antitumour immunity, suppress antigen presentation by DCs and CD4+ T cell function and promote intratumoural T cell exhaustion [40,41]. Exhausted CD8+ T cells have been demonstrated to lose effector functions and be unable to efficiently produce IL-2, TNF and IFN-γ [18]. Thus, these findings were in agreement with the expected negative clinical impact of locally immunosuppressive cells, such as Tregs and exhausted CD8+ T cells, promoting cancer growth and progression. However, there is a lack of evidence regarding the impact of the TME on prognosis. Our results showed that high Treg levels and exhausted CD8+ T cell levels were associated with a worse OS, suggesting that Tregs and exhausted CD8+ T cells might be prognostic biomarkers in *ALK*-rearranged NSCLC. This finding was consistent with a previous study showing that the increased infiltration of Tregs into core tumour regions is an independent predictor of worse OS in NSCLC [36]. Furthermore, we found that patients with low exhausted CD8+ T cell levels showed an association with a longer PFS in the PD-L1hi group. Such findings indicated that based on different TMEs, a subset of *ALK*-rearranged NSCLC patients with high PD-L1 expression may benefit from ALK-TKIs. The specific molecular mechanisms and signalling pathways that regulate the close interaction between PD-L1 and Tregs or exhausted CD8+ T cells within the TME were not explored in our study, and we need to further decipher the determinants of Treg and exhausted CD8+ T cell infiltration in patients with a high PD-L1 expression and *ALK*-rearranged NSCLC.

Additionally, our study found a strong association between exhausted CD8+ T cells and B cells. A previous study showed that B cells can secrete IL-10 and that blocking IL-10 can prevent and reverse T cell exhaustion [42]. Our findings imply that B cells promote CD8+ T cell exhaustion and dampen antitumour activity. However, the correlation analysis in this study was only based on the quantitative expression of biomarkers without considering the influence of location. Further analysis is needed to confirm this.

There are some limitations to our study. First, this was a retrospective study with a small sample size from a single institution. Second, the biopsy specimen tissue amount was small, and the tumour and the microenvironment were heterogeneous, and therefore the expression of biomarkers in the entire tumour tissue might be overestimated or underestimated. Third, only mIF was used to characterise the TME, and other tests, such as flow cytometry and single-cell RNA sequencing, should be conducted in future studies.

## 5. Conclusions

In conclusion, our study found an association between high PD-L1 expression and poor prognosis in *ALK*-rearranged NSCLC patients. Patients with high PD-L1 expression demonstrated an immunosuppressive TME, which was characterised by a higher Treg level and exhausted CD8+ T cell infiltration. The characteristics of the TME may help to identify patients who would greater benefit from ALK-TKIs.

## Figures and Tables

**Figure 1 biomolecules-13-00991-f001:**
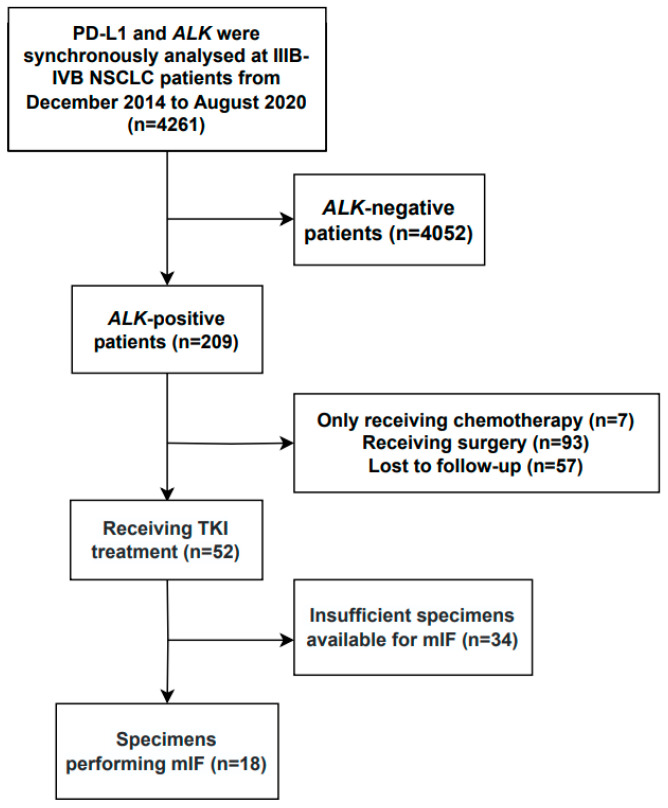
Flow chart of the study.

**Figure 2 biomolecules-13-00991-f002:**
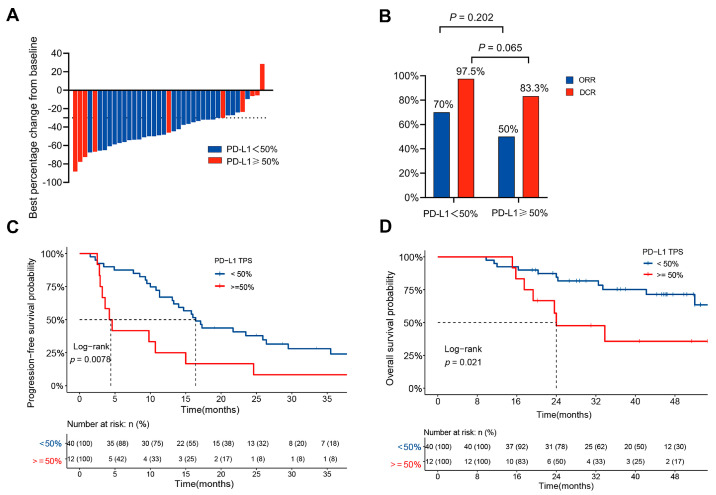
Survival analysis of 52 patients with *ALK*−rearranged NSCLC. (**A**) Maximum percentage reduction from the baseline sum of lesion diameters in 52 patients receiving ALK−TKI treatment. (**B**) Clinical response in patients receiving ALK−TKI treatments. (**C**,**D**) PFS (**C**) and OS (**D**) in patients with *ALK*−rearranged NSCLC stratified by PD-L1 TPS 50%.

**Figure 3 biomolecules-13-00991-f003:**
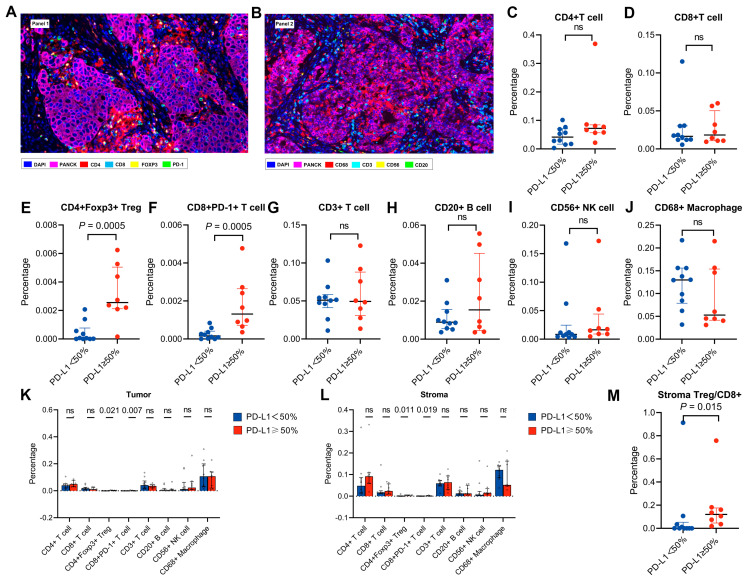
Features of the tumour microenvironment. (**A**,**B**) Representative multiplex immunofluorescence images of *ALK*−rearranged NSCLC tumour sections analysed for Panel 1 (**A**) and Panel 2 (**B**) markers. (**C**–**J**) Differences in the distribution of CD4+ T cells (**C**), CD8+ T cells (**D**), Tregs (**E**), exhausted CD8+ T cells (**F**), CD3+ T cells (**G**), B cells (**H**), NK cells (**I**) and macrophages (**J**) between PD−L1hi and PD−L1lo patients. (**K**,**L**) The distribution of the indicated immune cell infiltration in the stroma (**K**) and tumour region (**L**) between PD−L1hi and PD−L1lo patients. (**M**) Treg/CD8 ratio in PD−L1hi and PD−L1lo patients. ns, not significant (Mann-Whitney U test).

**Figure 4 biomolecules-13-00991-f004:**
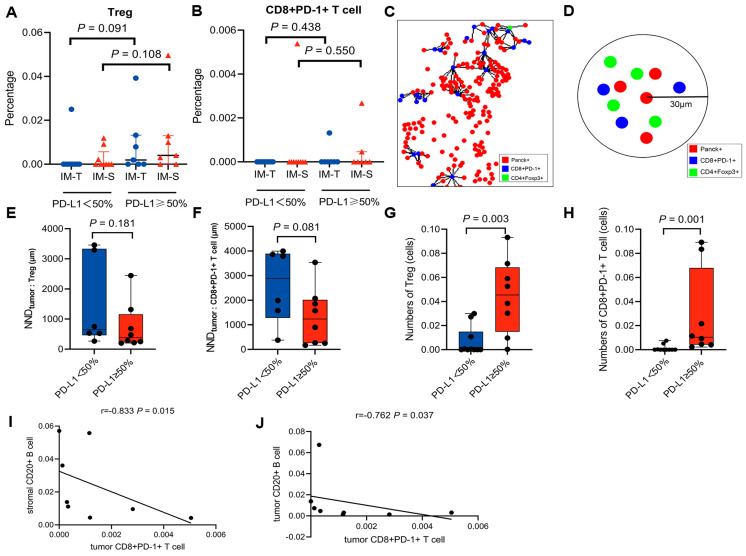
Location, spatial distribution and correlation analysis of Treg and exhausted CD8+ T cells. (**A**,**B**) Treg (**A**) and exhausted CD8+ T cell (**B**) percentages in the tumour and stroma of the invasive margin (IM) region. (**C**) Representative image showing the NND analysis, with the NND calculated from tumour cells to their nearest Treg and exhausted CD8+ T cells. (**D**) Cartoons depicting the number of cells within a 30 μm radius (tumour cells (red), Tregs (green), and exhausted CD8+ T cells (blue)). (**E**,**F**) Boxplots of the average NND between Tregs (**E**), exhausted CD8+ T cells (**F**) and tumour cells. (**G**,**H**) Boxplots showing the number of Tregs (**G**) and exhausted CD8+ T cells (**H**) within a 30 μm radius of tumour cells. (**I**,**J**) The correlation between exhausted CD8+ T cells and B cells in PD−L1hi patients.

**Figure 5 biomolecules-13-00991-f005:**
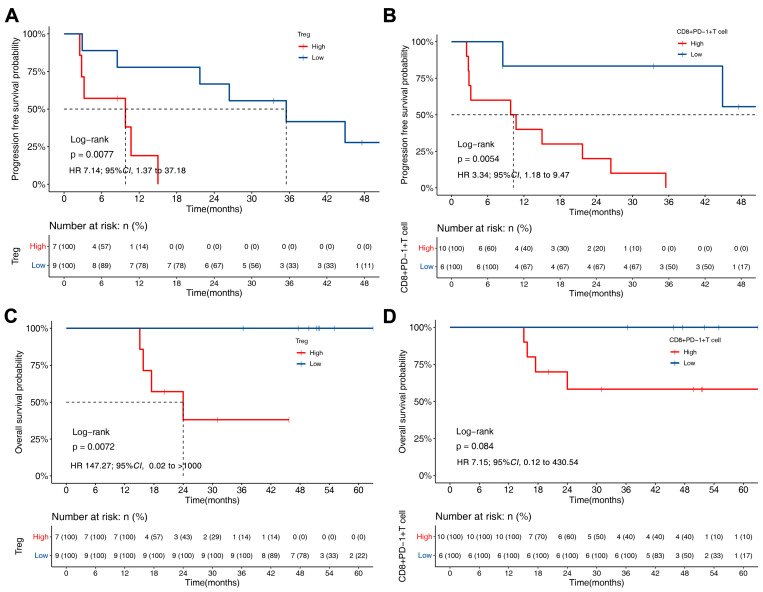
Kaplan–Meier analysis of PFS and OS according to the percentage of Treg and exhausted CD8+ T cells in patients treated with crizotinib. (**A**,**B**) The difference in PFS between patients with high and low levels of tumour-infiltrating Tregs (**A**) and exhausted CD8+ T cells (**B**). (**C**,**D**) The difference in OS between patients with high and low levels of tumour-infiltrating Tregs (**C**) and exhausted CD8+ T cells (**D**).

**Figure 6 biomolecules-13-00991-f006:**
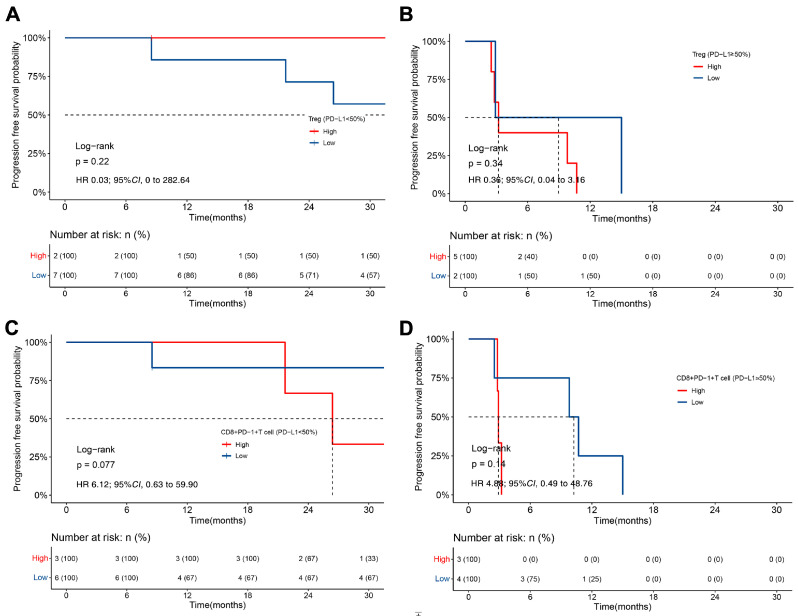
Kaplan–Meier analysis of PFS based on TME in patients treated with crizotinib. (**A**,**B**) The difference in PFS between patients with high and low levels of tumour-infiltrating Tregs in the PD-L1lo group (**A**) and PD-L1hi group (**B**). (**C**,**D**) The difference in PFS between patients with high and low levels of tumour-infiltrating exhausted CD8+ T cells in the PD-L1lo group (**C**) and PD-L1hi group (**D**).

**Table 1 biomolecules-13-00991-t001:** Clinicopathological characteristics of patients with *ALK*-rearranged NSCLC receiving ALK-TKIs.

Variable	Total
Age	
Median (25%, 75%)	51 (41, 61)
Sex, *n* (%)	
Male	21 (40%)
Female	31 (60%)
Smoking status, *n* (%)	
Non-smoker	38 (73%)
Former/current smoker	14 (27%)
Stage, *n* (%)	
IIIB–IIIC	3 (6%)
IV	49 (94%)
Histology, *n* (%)	
Adenocarcinoma	52 (100%)
ECOG PS, *n* (%)	
0	18 (35%)
1	27 (52%)
2	7 (13%)
Metastasis, *n* (%)	
Brain	18 (35%)
Bone	24 (46%)
Liver	11 (21%)
Adrenal glands	4 (8%)
Pleura	13 (25%)
Contralateral lung	18 (35%)
TKI, *n* (%)	
Crizotinib	42 (81%)
Alectinib	10 (19%)
PD-L1 TPS, *n* (%)	
0%	23 (44%)
1–49%	17 (33%)
≥50%	12 (23%)

ECOG PS, Eastern Cooperative Oncology Group performance status; PD-L1, programmed cell death-ligand 1; TKI, tyrosine kinase inhibitor.

**Table 2 biomolecules-13-00991-t002:** Baseline characteristics in the subgroups of patients with low and high PD-L1 TPS.

Variable	PD-L1 < 50% (*n* = 40)	PD-L1 ≥ 50% (*n* = 12)	*p* Value
Age			
<65 year	31 (77.5%)	10 (83.3%)	1.000
≥65 year	9 (22.5%)	2 (16.7%)	
Sex			
Male	16 (40.0%)	5 (41.7%)	1.000
Female	24 (60.0%)	7 (58.3%)	
Smoking status			
Never	28 (70.0%)	10 (83.3%)	0.475
Former/current	12 (30.0%)	2 (16.7%)	
Stage			
IIIB–IIIC	3 (7.5%)	0	1.000
IV	37 (92.5%)	12 (100.0%)	
ECOG PS			0.103
0	15 (37.5%)	3 (25.0%)	
1	22 (55.0%)	5 (41.7%)	
2	3 (7.5%)	4 (33.3%)	
Metastasis			
Brain	12 (30.0%)	6 (50.0%)	0.300
Bone	17 (42.5%)	7 (58.3%)	0.510
Liver	8 (20.0%)	3 (25.0%)	0.701
Adrenal glands	3 (7.5%)	1 (8.3%)	1.000
Pleura	10 (25.0%)	3 (25.0%)	1.000
Contralateral lung	11 (27.5%)	7 (58.3%)	0.082
TKI			
Crizotinib	32 (80.0%)	10 (83.3%)	1.000
Alectinib	8 (20.0%)	2 (16.7%)	

ECOG PS, Eastern Cooperative Oncology Group performance status; PD-L1, programmed cell death-ligand 1; TKI, tyrosine kinase inhibitor.

**Table 3 biomolecules-13-00991-t003:** Univariate and multivariate analyses for PFS.

Variable	Univariate Analysis		Multivariate Analysis	
	HR (95% CI)	*p* Value	HR (95% CI)	*p* Value
Age				
<65 year	Reference [1]			
≥65 year	1.342 (0.610, 2.952)	0.465		
Sex				
Male	Reference [1]			
Female	0.689 (0.367, 1.292)	0.245		
Smoking status				
Never	Reference [1]			
Former/current	1.723 (0.869, 3.418)	0.119		
Stage				
IIIB–IIIC	Reference [1]			
IV	1.319 (0.317, 5.495)	0.704		
ECOG PS				
0	Reference [1]			
1–2	1.087 (0.573, 2.061)	0.798		
Metastasis				
Brain	1.710 (0.902, 3.245)	0.100		
Bone	1.956 (1.045, 3.660)	0.036	2.123 (1.119, 4.028)	0.021
Liver	1.773 (0.862, 3.647)	0.120		
Adrenal glands	2.842 (0.993, 8.132)	0.051		
Pleura	1.004 (0.490, 2.057)	0.990		
Contralateral lung	1.733 (0.922, 3.255)	0.088		
PD-L1 expression				
<50%	Reference [1]			
≥50%	2.532 (1.247, 5.141)	0.010	2.792 (1.348, 5.785)	0.006
TKI				
Crizotinib	Reference [1]			
Alectinib	0.397 (0.155, 1.019)	0.055		

ECOG PS, Eastern Cooperative Oncology Group performance status; PD-L1, programmed cell death-ligand 1; TKI, tyrosine kinase inhibitor.

**Table 4 biomolecules-13-00991-t004:** Univariate and multivariate analyses for OS.

Variable	Univariate Analysis		Multivariate Analysis	
	HR (95% CI)	*p* Value	HR (95% CI)	*p* Value
Age				
<65 year	Reference [1]			
≥65 year	1.130 (0.371, 3.444)	0.830		
Sex				
Male	Reference [1]			
Female	0.859 (0.332, 2.219)	0.753		
Smoking status				
Never	Reference [1]			
Former/current	1.193 (0.425, 3.349)	0.738		
Stage				
IIIB–IIIC	Reference [1]			
IV	0.308 (0.068, 1.393)	0.126		
ECOG PS				
0	Reference [1]		Reference [1]	
1–2	5.870 (1.344, 25.644)	0.019	6.420 (1.449, 28.439)	0.014
Metastasis				
Brain	1.602 (0.616, 4.166)	0.334		
Bone	2.349 (0.901, 6.128)	0.081		
Liver	2.260 (0.835, 6.117)	0.109		
Adrenal glands	1.306 (0.300, 5.688)	0.722		
Pleura	0.891 (0.293, 2.710)	0.839		
Contralateral lung	1.864 (0.739, 4.699)	0.187		
PD-L1 expression				
<50%	Reference [1]		Reference [1]	
≥50%	2.945 (1.128, 7.691)	0.027	3.348 (1.231, 9.107)	0.018
TKI				
Crizotinib	Reference [1]			
Alectinib	0.275 (0.036, 2.076)	0.211		

ECOG PS, Eastern Cooperative Oncology Group performance status; PD-L1, programmed cell death-ligand 1; TKI, tyrosine kinase inhibitor.

## Data Availability

The datasets analysed during the current study are available from the corresponding author on reasonable request.

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
