# Peer review of "High PD-L1 Expression Correlates with an Immunosuppressive Tumour Immune Microenvironment and Worse Prognosis in ALK-Rearranged Non-Small Cell Lung Cancer"

_biomolecules, 2023, doi:10.3390/biom13060991_

Round 1
Reviewer 1 Report (Previous Reviewer 2)
The response to the point is understandable. However, regarding the multivariate analysis for survival, it appears that all of PS, PS 1, and PS>2 are in the multivariate analysis. This is obviously not acceptable, and the description or analysis needs to be revised.
The abstract also states that PS ≥ 2 is related to survival, but this statement deviates from Table 4, which shows a difference even for PS 1.
Regarding the PFS curves in Figures 5 and 6, it does not seem meaningful to compare PFS in mixed crizotinib and alectinib cases. Is it correct to say that these are for Crizotinib-treated cases only? If so, this should be stated.
Author Response
Please see the attachment.

Reviewer 2 Report (Previous Reviewer 1)
No further comments
Author Response
Thank you for reviewing my article. There are no comments that need to be revised.
Round 2
Reviewer 1 Report (Previous Reviewer 2)
The responses to the previous points are understandable.
There are no new points regarding scientific and data aspects.
However, re-checking of English by native speakers is necessary.
Also, there are some errors in the description. For example, in the newly added Table S4, the data provided are only for patients treated with crizotinib, but at the end of the text there is a note that reads “Clinical outcomes of 18 patients with ALK-positive NSCLC treated with ALK TKIs”, which is not correct.
And in abstract line 16, “pretreatment” should be “pretreated”.
Throughout reviews, there are many inconsistencies and errors in the descriptions. Careful attention and action are required before publication.
re-checking of English by native speakers is necessary.
Author Response
Please see the attachment.

This manuscript is a resubmission of an earlier submission. The following is a list of the peer review reports and author responses from that submission.
Round 1
Reviewer 1 Report
This is an excellent work based on a very large cohort size. I just have couple of minor comments:
1) ALK rearrangement analysis methodology should be mentioned.
2) Hazard ratio and Confidence Interval should be mentioned for figure 5 &6.
3) Did any patients receive immunotherapy in this coghort, if so, how were the outcomes?
4) Please discuss your finding in the context of the clinical trials (IMPOWER 130, 132 or KEYNOTE407 etc).
Reviewer 2 Report
This report, which retrospectively evaluated PD-L1 tumor expression and tumor microenvironment in ALK-positive lung cancer, is a valuable report for ALK-positive lung cancer, which, as the authors point out, is poorly explored.
However, there are important concerns with this report.
1. The main point of this report is the comparison of PD-L1 high and non-high expression. However, the background of this PD-L1 high and low expression is not shown. First of all, the background of the two groups should be clearly indicated in the Table to show how much differences are there between the two groups. It is especially important whether the ALK inhibitor used was crizotinib or alectinib. Also, more information is needed on prior therapy. Furthermore, no information on PS is found in this manuscrpt. As the most important prognostic factor, PS must be clearly indicated and need to be contained in statistical analysis.
2.  For PFS comparisons, it is unacceptable to ignore differences in drug, since the PFS of different crizotinib and alectinib is more than 3-fold different. The PFS analysis must include only cases using crizotinib or alectinib.
3. Explanation is needed for the reason why PD-L1high has a worse prognosis than low, for the difference between ALK-positive and ALK-negative lung cancer (difference from the previously reported data) regarding the results of TME evaluation. Explanation for why PD-1+CD8+T cells and regT cells are associated with the efficacy (PFS) of ALK inhibitors is also required in discussion session.
4. The reasons for the NND analysis, the significance of the results, and how the results presented here differ from similar results in the past, especially the comparison between ALK-positive and negative lung cancers, should be discussed. In addition, it has been shown that there is a correlation between CD8+ T cells and B cells, and the significance of this should be evaluated and discussed in the text.
Round 2
Reviewer 2 Report
There are major problems with the statistical analysis.
First of all, it is unclear how they decided which items to include in the multivariate analysis.
In Table 3, PS has p=0.072, but is not included in the multivariate analysis. Alectinib is also not included, although p=0.058. However, contralateral lung was included at P=0.086.
In Table 4, PS is newly included in the multivariate analysis with P=0.017. However, why are the HR and P values for bone metastases unchanged?
Together with the fact that there is no description of the multivariate analysis method, the statistical results of this study are not reliable.
In the Discussion session, the addition of a new explanation is commendable, but the description on lines 330-335 of 12P is incomprehensible.
What the author indicates should be a discussion on the difference in TME between high and low PD-L1 levels and its impact on prognosis, and not related to the involvement of PD-L1 expression in resistance to TKIs, is it not?